# Versatile Triad Alliance: Bile Acid, Taurine and Microbiota

**DOI:** 10.3390/cells11152337

**Published:** 2022-07-29

**Authors:** Kalina Duszka

**Affiliations:** Department of Nutritional Sciences, University of Vienna, 1090 Vienna, Austria; kalina.duszka@univie.ac.at

**Keywords:** taurine, bile acids, microbiota

## Abstract

Taurine is the most abundant free amino acid in the body, and is mainly derived from the diet, but can also be produced endogenously from cysteine. It plays multiple essential roles in the body, including development, energy production, osmoregulation, prevention of oxidative stress, and inflammation. Taurine is also crucial as a molecule used to conjugate bile acids (BAs). In the gastrointestinal tract, BAs deconjugation by enteric bacteria results in high levels of unconjugated BAs and free taurine. Depending on conjugation status and other bacterial modifications, BAs constitute a pool of related but highly diverse molecules, each with different properties concerning solubility and toxicity, capacity to activate or inhibit receptors of BAs, and direct and indirect impact on microbiota and the host, whereas free taurine has a largely protective impact on the host, serves as a source of energy for microbiota, regulates bacterial colonization and defends from pathogens. Several remarkable examples of the interaction between taurine and gut microbiota have recently been described. This review will introduce the necessary background information and lay out the latest discoveries in the interaction of the co-reliant triad of BAs, taurine, and microbiota.

## 1. Introduction

Bacteria, archaea, viruses, and fungi sum up to trillions of microorganisms inhabiting digestive systems. Gut microbiota, owing to its complexity and diversity, but also therapeutic applicability, benefits, and detriments, has received considerable interest from the scientific community and public. What we have learned to date, is that the composition of microbial communities depends on multiple factors, including gender, birth mode, disease history, or geographical location. However, the composition is anything but static. It evolves with age, exposure to environmental factors (pollutants, habitat, seasons), pathogens, drugs (particularly antibiotics and proton pump inhibitors), lifestyle changes (e.g., physical activity), and most of all, the diet [1,2]. Moreover, it also fluctuates according to circadian rhythm [3,4]. In addition to the external factors, microbiota-generated metabolites influence the gut flora, modulating their composition, restricting population size, and protecting from pathogens by producing antibacterial factors, e.g., secondary bile acids (BAs). BAs, as well as other signaling molecules derived from or triggered by bacteria, such as short-chain fatty acids (SCFA), neurotransmitters, hormones, ligands for host’s receptors, and modulators of metabolism (e.g., glucagon-like-peptide-1 (GLP-1)); all have developed over centuries of symbiosis, serving in constant communication between the resident gut bacteria and the host. Consequently, the microbiota is vital to the host’s health, including immune, metabolic, and neurobehavioral aspects. Dysbiosis and reduced bacterial diversity correlate with multiple diseases, including inflammatory bowel disease, atopic eczema, coeliac disease, obesity, arterial stiffness, lupus, liver diseases, multiple sclerosis, Parkinson’s disease, psoriasis, rheumatoid arthritis, and type 1 and 2 diabetes [2,5,6,7]. The importance, signaling, and health-relevance of gut bacteria have been reviewed multiple times [1,6,8,9,10] and will not be the scope of the current article. This review will, in turn, focus on the role of taurine in the context of gut bacteria. Firstly, taurine with its various functions will be introduced. Since microbiota-driven deconjugation of taurine-conjugated BAs is the primary source of endogenous taurine in the GI tract, BAs’ modifications and signaling will be described. Finally, the newest developments in research concerning taurine in the context of the gastrointestinal (GI) tract and its interaction with microbiota will be discussed.

## 2. Taurine

Taurine or 2-aminoethylsulfonic acid is a β-amino acid containing a sulfonate instead of a carboxylic group. Taurine is present in high intracellular concentrations in the human brain, retina, heart, skeletal muscles, and leukocytes, and is vital for organ development [11]. Taurine primarily exists in an organism in free form, as it is not used for protein synthesis, and is referred to as a conditionally essential amino acid. Therefore, next to aspartate in the human liver, it constitutes the most abundant free amino acid in mammalian tissues [12,13,14]. Based on the expression levels of synthesizing enzymes, the liver is the leading site of endogenous taurine generation. However, smaller amounts are also produced in neurons, glial cells, pancreas, ovary, heart, kidney, and muscle [12,15]. Taurine is synthesized through the cysteine sulfinic acid or transsulfuration pathway involving cysteine oxidation by cysteine dioxygenase (CDO), subsequent decarboxylation by cysteine sulfonate decarboxylase (CSD), and oxidation of the generated hypotaurine by hypotaurine dioxygenase. The alternative pathway involves cysteamine dioxygenase (ADO) that converts the coenzyme A-derived cysteamine to hypotaurine and further to taurine [16,17,18,19]. The rate of taurine biosynthesis varies between individuals in relation to the nutritional state, the amount of protein intake, and the availability of cysteine as the substrate [12,20,21]. In turn, the availability of cysteine is highly dependent on the metabolic equilibrium between homocysteine and methionine via folic acid, vitamin B12, and the efficiency of the methyltetrahydrofolate reductase. However, a certain amount of taurine is preferably taken in with food, mainly by human infants, carnivores, and, to a minor extent, omnivores [12,19]. Taurine is contained in high quantities in shellfish, particularly mussels, scallops, clams, and dark meat of chickens and turkeys. In general, taurine is not found in the majority of plant products such as vegetables, nuts or seeds, fruits, and legumes [22]. Consequently, vegans have lower levels of taurine [23,24]. However, medicinal plants such as red goji fruit (*Lycium barbarum* L.) are a rich source of taurine [25]. Importantly, cooking does not reduce the levels of taurine. Average dietary intake of taurine has been estimated between 40 and 400 mg for non-vegetarians depending on the reporting source, with consumption up to 3000 mg considered safe [26,27,28].

Taurine plays a role in a broad spectrum of functions in multiple organs. Due to the high tissue concentration, taurine works as an osmolyte [29]. Its cellular efflux via volume-dependent or volume-independent pathways is tightly regulated at the transcriptional and post-transcriptional levels. Taurine levels are ensured by a specific, active sodium and chloride ion-dependent transporter (TauT; encoded by the SLC6A6 gene) that concentrates taurine inside the cells against gradients [30,31,32]. Taurine is needed for membrane stabilization, has cytoprotective effects, and maintains calcium homeostasis and signaling by affecting ion channel function [12,33,34]. Scarcity of taurine impairs growth [35], fertility [36,37,38,39,40], triggers immune deficiency [41], muscle atrophy, cardiomyopathy, and renal and pancreatic dysfunction [18,42,43]. In the male reproductive system, taurine is detected at relatively high levels [44,45]. It is necessary for fertility as it affects sperm osmoadaptation [36], capacitation [37], and motility [38,39,40]. Considering metabolism, taurine is essential for skeletal and heart muscle function [33,46]. It regulates neuroendocrine functions of pancreatic β-cells [47], liver glucose levels [48], leptin [49], and insulin signaling [49,50]. Taurine supports the function of the central nervous system and eyes. It is considered one of the most critical amino acids in brain development, including proliferation and development of neuronal cells, as well as protecting neurons from excitotoxicity induced by excitatory amino acids [51]. Therefore, due to its capacity to modulate hyperreactivity, hyperpolarize, and inhibit neuron firing, it is applicable in epilepsy [52]. It serves as an agonist of GABA_A_ receptors and, through them, exerts its neuronal inhibitory, anxiolytic, and calming effect [53,54]. Consequently, taurine promotes emotional learning ability, memory, and cognitive performance [55,56]. It plays a role in the prevention of nerve system-related abnormalities, including depression [57], epileptic seizures [58], retinal and tapetum degeneration [35,59]. It protects against neurodegeneration [60,61], as exemplified in disease models for Parkinson’s [62], Alzheimer’s [63], and Huntington’s [64].

Taurine prevents inflammatory damage by acting on nuclear factor ‘kappa-light-chain-enhancer’ of activated B-cells (NFkB)/cyclooxygenase (COX), p38 mitogen-activated protein kinases (MAPK)—c-jun N-terminal kinase-dependent (JNK), and myeloid differentiation primary response 88 (MyD88) signaling mediated by toll-like receptor 4 (TLR-4). It reduces inducible nitric oxide synthase (iNOS), C-reactive protein (CRP), and is the marker of lipid peroxidation thiobarbituric acid reactive substances (TBARS) [65,66,67,68]. However, the primary benefits of taurine supplementation are derived from its antioxidative activities [30,61,69,70,71]. Taurine itself prevents oxidative damage by increasing enzymatic and nonenzymatic antioxidants. Whereas taurine chloramine inhibits the production of superoxide anion (O^2–^) and nitric oxide (NO) [72,73]. In parallel, it increases the expressions of cytoprotective antioxidant proteins, such as heme oxygenase 1 (HO-1), peroxiredoxin (PRX), and thioredoxin (TRX), in macrophages [74]. It also inhibits reactive oxygen species by Kelch-like ECH-associated protein 1 (Keap-1)/nuclear factor erythroid-2-related factor (Nrf2)/heme oxygenase-1 (HO-1) pathway [75]. Taurine has been shown to alleviate liver damage in models of nonalcoholic fatty liver (NAFL) and alcoholic liver disease. One study reported enhanced liver antioxidant capacities via glutathione (GSH), Trolox equivalent antioxidant capacity (TEAC), superoxide dismutase (SOD), and catalase (CAT), decreased lipid peroxidation and malondialdehyde (MDA) levels [76]. At the same time, the other report claimed a positive impact of taurine by opposite means: increasing MDA, reducing SOD, GSH peroxidase (GSH-Px), total antioxidant capacity (T-AOC), COX, and NADH dehydrogenase (ND) [77].

Due to the existence of the taurine transporter system across the mitochondrial membranes [78], taurine is preferentially localized in the mitochondria, compared with the cytosol [79]. Consequently, taurine is found in very high concentrations (15–20 µmol/g) in oxidative tissues, which are characterized by a high number of mitochondria, but in lower concentrations in glycolytic tissues (1–3 µmol/g) [78,80,81]. Notably, one of the ways by which taurine stabilizes mitochondrial oxidation and metabolic function is by acting as a mitochondrial matrix buffer [78,82]. However, it can also be conjugated in the mitochondria of extra-hepatic tissues to 5-taurinomethyl uridine with the wobble position uridine of tRNA^Leu(UUR)^. This process enhances the interaction between the modified uridine and guanine and modulates the synthesis of mitochondrial proteins [83]. Biochemically, the modification enhances the binding of the UUG codon to the taurine-modified AAU anticodon of tRNA^Leu(UUR)^, facilitating UUG decoding. Upon blockage of the formation of 5-taurinomethyluridine-tRNA^Leu(UUR)^, the biosynthesis of some mitochondrial proteins declines. Consequently, respiratory function falls, ATP generation decreases, and the generation of oxidants by the respiratory chain increases [84,85].

Furthermore, in mitochondria, taurine mediates energy generation, thermogenesis, and fatty acid oxidation [86,87]. In mice, taurine enhances adaptive thermogenesis and the browning of inguinal white adipose tissue by phosphorylating and, therefore, activating AMP-activated protein kinases (AMPK) and subsequently stimulating peroxisome proliferator-activated receptor γ coactivator 1-α (*Pgc1α*) and uncoupling protein 1 (UCP1) [76,88]. Corresponding stimulation occurs in human tissues. Taurine supports exercise-driven lipid metabolism and ensures mitochondrial function by regulating essential mitochondrial genes: cell death-inducing DFFA-like effector A (*Cidea*), *Pgc1α*, and PR domain containing 16 (*Prdm16*), *Ucp1*, and *Ucp2* [89,90,91]. The role of taurine in thermogenesis and energy expenditure relies on its capacity to stimulate fatty acid oxidation, which also occurs via the AMPK-PPARα pathway and involves multiple factors, including arnitine palmitoyltransferase 1-α (CPT1α), lipoprotein lipase (LPL), aconitase 1 (ACO1), aconitase 2 (ACO2), hormone-sensitive lipase (HSL), acyl-CoA oxidase-1 (ACOX1), cluster of differentiation 36 (CD36), and peroxisome proliferator-activated receptor-γ (PPARγ) [89]. In the case of PPARγ, taurine has been shown to increase its mRNA and protein levels [92]. At the same time, activation of PPARγ in the placenta correlates with the expression of TauT [93]. Taurine also inhibits lipid synthesis [94], reduces cholesterol and triglycerides (TG) by acting via the Sirtuin 1 (Sirt1)-APMK-sterol regulatory-element binding proteins (SREBP) 1c pathway and affecting acetyl CoA carboxylase (ACC), stearoyl-CoA desaturase 1 (SCD-1), PPARα, acyl-CoA oxidase (ACO), MTP, CPT-1a, and fatty acid synthase (FAS) [76,91,94,95,96,97]. Consequently, taurine supplementation results in increased body temperature [91], reduced body weight [95] or body weight gain in obesity models [88,96], alleviated high-fat diet (HFD)-induced obesity, improvement in the lipid profile, metabolic risk factors and insulin sensitivity [88,98]. Therefore, taurine may find application in the therapy of diabetes [60,99].

Taurine supplementation has been proven to reduce plasma cholesterol in diet-induced animal models of hypercholesterolemia, including rats [94,100,101], mice [102], hamsters [103], and quails [104]. Mechanistically, taurine inhibits cholesterol synthesis by AMPK-SREBP2-3-hydroxyl-3-methylglutaryl-CoA reductase (HMGCR). It also serves as a direct ligand of liver X receptor α (LXRα) and elevates the hepatic low-density lipoprotein (LDL) receptor [76,103,105], regulating the reverse cholesterol transport [106]. Upon accumulation in the liver, taurine supports the conversion of cholesterol into BAs via enhancement of the cholesterol 7α-hydroxylase (CYP7A1) enzyme [94,103,105,107,108,109]. In addition to regulating BAs synthesis, taurine also serves as a conjugation substrate for BAs, substantially changing their properties [110].

## 3. BAs

### 3.1. BAs Production

BAs are a group of cholesterol-derivatives of diverse structures and with amphiphilic properties. The primary BAs are chenodeoxycholic acid (CDCA) and cholic acid (CA) in humans, while rodents produce additional muricholic acids (αMCA and βMCA) as well as ursodeoxycholic acid (UDCA). Interestingly, in humans, UDCA constitutes secondary BA. A complex network of enzymes and factors controls BA synthesis and circulation (Figure 1). The synthesis of BAs involves 17 enzymes situated in the endoplasmic reticulum, mitochondria, cytosol, and peroxisomes of hepatocytes. BA synthesis can be accomplished via two distinct pathways. The classical pathway, also known as neutral, is initiated by the rate-limiting enzyme CYP7A1, which 7α-hydroxylates cholesterol followed by sterol 12α-hydroxylase (CYP8B1) and sterol 27-hydroxylase (CYP27A1), leading to the generation of CDCA and CA, respectively [111]. The alternative pathway, also referred to as acidic, is initiated by sterol-27-hydroxylase (CYP27A1), cholesterol 24-hydroxylase (CYP46A1), and cholesterol 25-hydroxylase (CYP3A4), followed by oxysterol 7α-hydroxylase (CYP7B1), and predominantly generates CDCA [112]. In the human liver, the classic pathway is the predominant one and contributes to 90% of hepatic BA production. Whereas in newborns, before the classic pathway is established, the alternative pathway generates the majority of BAs. In mice, both pathways are equally active [113,114].

### 3.2. BAs Conjugation

Once produced, BAs may undergo conjugation (N-acyl amidation) of C-24 to create bile salts. Conjugation with taurine or glycine increases polarity and water solubility and, therefore, affects BAs’ mobility such that passive transport is limited and transporters control BA reabsorption. Conjugation changes BAs’ properties decreasing their PKa value to approximately five, thus making them almost fully ionized at physiological pH and soluble over a wide range of ionic strengths, calcium concentrations, and pH values [115,116]. Micelles of conjugated BAs can bind calcium ions and thereby reduce calcium precipitation and the formation of gallstones [117]. Most importantly, due to their dual nature, conjugated BAs can act as efficient emulsifiers aiding lipid absorption. In mice, 95% of BAs are conjugated to taurine, while in humans, 70% are conjugated to glycine and 30% to taurine. However, diet affects these values. As a result, the ratio of glycoconjugates to tauroconjugates in humans may shift to as high as 9:1 in rural African women or as low as 0.1:1 upon taurine supplementation [118,119]. HFD increases levels of taurine-conjugated BAs [120]. In comparison, intraduodenal administration of only 250 mg of taurine in humans increases taurine conjugation of BAs by 2.5% to 10% within 2.5 h in seven out of ten subjects [118]. Therefore, taurine supplementation in infants’ formula, which increases intestinal fat absorption, acts most likely by increasing the pool of taurine conjugated, emulsifying BAs [121]. The two-step reaction of taurine conjugation is catalyzed by BA CoA-ligase (BAL) and BA CoA:amino acid N-acyltransferase (BAT) enzymes [122]. In addition to taurine and glycine, BAs can also be amidated with other amino acids, including leucine or lysine, but these conjugates are rapidly hydrolyzed by pancreatic carboxypeptidases [123].

### 3.3. BAs Transport

BAs produced by hepatocytes are secreted actively across the canalicular membrane by several transporters, including the bile salt export pump (BSEP) and multidrug resistance-associated protein 2 (MRP2). Collected into a canalicular network, BAs are drained into bile ducts that merge to form hepatic ducts. BAs can then be concentrated in the gallbladder or secreted directly to the duodenum upon cholecystokinin-stimulated gallbladder contractions following meal ingestion [124]. The primary role of BAs is to aid lipid and lipid-soluble vitamin solubilization, digestion, and absorption [125]. Depending on the diet, demands for BAs levels change, e.g., emulsification of long-chain saturated fats requires more BAs than shorter fatty acids [126,127]. Furthermore, fat digestion may become less efficient with increasing levels of ingested fat [128,129,130]. In addition to fat, BAs support glucose management. One of the mechanisms includes stimulation of GLP-1 and peptide-YY (PYY) secretion from enteroendocrine L cells [131]. Interestingly, BAs are also secreted during caloric restriction [132]. In that scenario, it is likely that motilin, of which levels are increased during caloric restriction, stimulates gallbladder emptying [133]. The role of BAs in the context of caloric restriction is unclear, however, it is expected to enhance nutrient uptake from potentially upcoming meals [132].

Upon secretion, BAs travel along the GI tract, absorbed by passive diffusion along the entire gut until the distal small intestine, where BAs are actively reabsorbed via the ileal BA transporter (IBAT, also known as apical sodium-dependent BA transporter (ASBT)) and secreted with organic solute transporter α-β (OSTα-OSTβ) on the basolateral side of epithelial cells. Approximately 90–95% of BAs are recirculated via the portal bloodstream and rapidly taken up by hepatocytes with the help of the sodium-dependent taurocholate co-transporting peptide (NTCP) or sodium-independent organic anion-transporting polypeptides (OATPs), OATP1B1 and OATP1B3. The reabsorbed BAs are subsequently reconjugated and resecreted into bile. The process is referred to as enterohepatic circulation and occurs in humans about six times per day [125,134,135,136,137,138,139,140]. BAs escaping enterohepatic circulation are transported to the colon, undergo more advanced biotransformation, and are taken up by passive absorption through the colonic mucosa or secreted with feces.

### 3.4. BAs Signaling

Upon circulation, BAs encounter their receptors in the GI tract and liver. The most essential of them, farnesoid X receptor (FXR), is a nuclear receptor. CDCA serves as the most potent ligand of FXR, followed by CA, deoxycholic acid (DCA), and lithocholic acid (LCA) [141,142,143]. UDCA does not activate but instead inhibits FXR [143,144], and taurine-conjugated MCAs (TαMCA and TβMCA) are competitive FXR antagonists. Activation of FXR and heterodimerization with its partner retinoid X receptor (RXR), stimulates the production of its target protein fibroblast growth factor 19 (FGF19) (FGF15 in mice) in the distal ileum. FGF15/19 travels with the portal blood to the liver, where it binds the fibroblast growth factor receptor 4 (FGFR4)/β-klotho heterodimer complex. It thus initiates signaling via the JNK1/2 and extracellular signal-regulated protein kinase (ERK1/2) pathway. Consequently, the information on BA abundance is communicated, and the activity of Cyp7a1 is inhibited [145,146,147]. However, FGF15/19 signaling has a myriad of other outcomes, including regulation of glucose metabolism, lipogenesis, and metabolic rate [148]. It also increases the stability of the hepatic small heterodimer partner (SHP) by inhibiting its proteasomal degradation [149]. SHP interacts with various nuclear receptors and blocks their transactivation [150]. *Fxr* is one of SHP’s direct target as well as its transcriptional repressor. The induction of *Shp* by FXR results in a cascade of inhibitory actions, including liver receptor homolog-1 (LRH-1) and resulting in the suppression of *Cyp7a1* expression and de novo BA synthesis [151,152]. Another pathway affected by FXR-induced SHP involves repression of the ability of hepatic nuclear factor 4α (HNF-4α) to induce CYP8B1 expression [153]. Interestingly, ileal FXR suppresses *Cyp7a1* more potently than the hepatic one, whereas hepatic FXR impacts *Cyp8b1* and thus the alternative pathway [145,154,155]. In addition to establishing the negative feedback loop, FXR also modulates BA metabolism, uptake from the intestine, and circulation, by regulating the expression of ileal BA binding protein (I-BABP) [156], IBAT [157,158,159], OSTα-OSTβ [140], hepatocellular uptake transporter OATP8 [160], MRP2 [161], BA-CoA synthetase (BACS), and BAT [162].

Since FXR takes input from microbiota, GF mice have down-regulated levels of IBAT and consequently decreased BAs uptake and elevated residual levels of BAs in the cecum. This regulation happens even though GF mice have ca. 71% increased BA pool size compared with conventional animals [157,158,159].

In addition to BA, FXR regulates glucose homeostasis [163,164,165,166]. It represses hepatic gluconeogenesis [165], interferes with glycolysis [165], stimulates glycogen storage [167], glucose transporter 4 (GLUT-4) [168], and insulin sensitivity [169]. Regarding lipid metabolism, FXR induces the expression of genes involved in lipoprotein metabolism/clearance, represses de novo lipogenesis, and increases β-oxidation [163,164,167,170]. FXR also takes part in inflammation as it reduces the expression of MCP1, IL1β, IL2, IL6, TNFα, and IFNγ genes as well as leukotrienes production, NKT cell, and NFkB activation [171,172,173].

BAs also serve as ligands for another important BAs receptor, Takeda G protein-coupled receptor 5 (TGR5 or GPBAR1) [174,175,176,177]. TGR5 is a ubiquitously expressed cell surface-located receptor. It recognizes both conjugated and free BAs with a preference for LCA > DCA > CDCA > CA [177]. Taurine conjugation of BAs increases the affinity for TGR5 as compared with unconjugated BAs; conversely, conjugation with glycine reduces the TGR5 affinity [174]. The activation of TGR5 increases intracellular cAMP and further activates TRPA1 and Epac pathways, protein kinase B (AKT), and downstream mammalian target of rapamycin (mTOR), NO, and GSK3B, also by inhibiting RhoA, STAT3, NFkb, or depending on the context, adjusting the activity of ERK1/2. Overall, the signaling results in an array of physiological outcomes, including differentiation, proliferation, survival, and metabolism. TGR5 activity improves liver function, reduces hepatic inflammation, steatosis, and fibrosis, contributes to hepatic glucose metabolism and insulin signaling, and maintains metabolic homeostasis [178,179]. One of the most vital aspects of TGR5 signaling involves dose-dependent elevation of GLP-1 secretion from enteroendocrine L cells, which improves glucose homeostasis [180,181]. Conversely, FXR function in the colon has been linked to the inhibition of GLP-1 secretion [182,183]. However, activation of FXR has also been reported to induce TGR5 to stimulate GLP-1 secretion [184]. TGR5 also induces metabolic responses in other tissues, including thermogenesis [185] and browning of white adipose tissue [166], increased energy expenditure in brown adipose tissue [181], and increased insulin production by pancreatic β cells [186]. However, its capacity to stimulate gallbladder filling and contribute to gallstone formation is impeding its application as a therapeutic target [187].

### 3.5. Microbiota and BAs

#### 3.5.1. The Impact of BAs on Microbiota

Circulating between organs, BAs play a role in various physiological processes through their combined signaling, detergent, and antimicrobial activities. BAs, next to antibacterial peptides, proteolytic enzymes, or rapid transit times, are one of the mechanisms via which the host suppresses microbial colonization and prevents excessive proliferation in the GI tract. Therefore, cases of reduced BA levels or obstruction of bile flow are associated with bacterial overgrowth, inflammation, and mucosal injury [188,189,190]. Various types of BAs’ modifications change their impact on microbiota. BAs, particularly the hydrophobic ones such as CDCA, LCA, and DCA, act as potent antimicrobial agents [191]. Therefore, the increase in hydrophobicity of DCA gives it much more potent detergent properties than its precursor CA [191]. BAs primarily exert their toxic effects on cell membranes by affecting the phospholipids and membrane proteins, which require a relatively high concentration of BAs and can be potentiated or attenuated depending on environmental conditions. For instance, *Listeria monocytogenes* grown under 40% CO_2_:60% N_2_ or 100% CO_2_ are highly susceptible to BAs. However, bacteria grown under air or 100% N_2_ recover following exposure to usually lethal concentrations of BAs [192]. *Lactobacillus acidophilus* cell membrane shows higher glycolipid/phospholipid and C18:2/C18:0 ratio as well as an increase in C16:0 and decrease in C19-cyc fatty acids when grown at 25 °C compared with 37 °C. These differences result in enhanced lipid membrane stability and more resistance to BAs-triggered permeability [193]. Finally, the bile-tolerant mutants of *Lactobacillus acidophilus* are characterized by altered cell wall fatty acid profiles, among other differences [194].

In addition to the impact on the cell membrane, BAs induce secondary structure formation in RNA, induce DNA damage, and activate enzymes involved in DNA repair in both bacterial and mammalian cells [195,196,197,198,199,200]. The detergent actions of bile may also alter the conformation of proteins resulting in their misfolding or denaturation. Furthermore, BAs may cause oxidative stress by generating free radicals, triggering an oxidative stress response in *Escherichia coli* [195], or acting as a chelating agent, reducing intracellular calcium and iron concentrations [117,201]. In addition to the direct antimicrobial effects, BAs execute indirect effects through activities of FXR. FXR stimulates the expression of genes involved in enteroprotection, including antimicrobial agents (e.g., iNOS and IL-18) [202] and angiogenin (Ang1), which acts as a part of the acute phase response to infection and has potent antibacterial and antimycotic actions [202]. This way, FXR inhibits bacterial overgrowth and mucosal injury in the ileum caused by bile duct ligation [202].

Gram-positive bacteria are much more sensitive to low concentrations of BAs. In contrast, gram-negative bacteria are more resistant; therefore, BAs can be applied in their selective enrichment in cultures [203]. Consequently, CA administration increased the ratio of mainly gram-positive *Firmicutes* to gram-negative *Bacteroidetes* [204]. A similar impact on microbiota has been observed as a consequence of HFD-feeding, which is known for triggering increased BA levels [205,206] and is also the case for obesity [207,208]. Importantly, bacteria can be trained and adapt to the presence of BAs [199,209,210,211], but conditioning the microbiome by applying HFD can amplify host energy harvest and inflammation [208,212,213]. High-saturated fat diet-triggered increase in BAs production may result in enriching for BAs-tolerant taxa in the families *Ruminococcaceae* and *Lachnospiraceae*, as well as the genera *Bacteroides* and *Bilophila*, that have previously been associated with host inflammation [120,212,214]. Several strains, including *Campylobacter*, *Salmonellae*, and *E. coli*, are also considered bile-resistant and have the ability to colonize the gallbladder [215,216,217,218,219,220,221,222,223,224,225]. Overall, high levels and different types of BAs can select for BA-resistant gut microbiota while inhibiting the growth of more sensitive organisms.

Synthesis and conjugation of taurine are also under microbial regulation. In the liver of germ-free (GF) mice, the gene expression of *TauT* and a rate-limiting enzyme for taurine synthesis *Csd* are increased, likely to compensate for drastically reduced taurine levels. Additionally, the levels of *Bacs* and its upstream regulator *Hnf4a1* are downregulated in the livers of GF mice, but *Bat* is not affected [157]. Therefore, it may seem surprising that in the liver of GF mice, taurine-conjugated BAs TCA and TβMCA are still the most prevalent, similar to the liver of conventional mice. However, in contrast to the distal small intestine of conventional mice, which is almost depleted of taurine-conjugated BAs, in GF mice, taurine-conjugated BAs are prevalent [157]. Therefore, the relatively high levels of taurine-conjugated BAs in the GF mice stem from the absence of BAs deconjugation.

#### 3.5.2. The Impact of Bacteria on BAs

The microbiota plays a vital role in the health and proper function of the GI tract and the whole body. As such, it also takes part in regulating BAs generation and metabolism, partly by affecting the activity of BA synthesis pathways. GF mice experiments have proven that gut bacteria, in general, reduce liver expression of *Cyp7a1*, *Cyp7b1*, and *Cyp27a1*, as well as enzymatic activity of CYP7A1 [157]. The process of microbial CYP7A1 inhibition involves upregulation of FXR-FGF15 signaling, and the subsequent reduction in BAs concentrations in the gut promotes microbial colonization [189,226]. On the other hand, gut microbiota modulation induced by VSL#3 probiotics promote ileal BA deconjugation with subsequent fecal BA excretion and induce hepatic BA neosynthesis via downregulation of the gut–liver FXR-FGF15 axis [227]. Therefore, the stimulation or inhibition of BAs’ synthesis is bacterial strain specific.

Concerning BAs, the central role of microbiota is connected with its capacity to submit primary BAs to a wide variety of modifications, including dihydroxylation, epimerization, oxidation, reduction, hydroxylation, and dihydroxylation, thus generating secondary BAs. The main reactions catalyzed by the gut bacteria involve the 7α-dehydroxylase-driven conversion of CA to DCA and CDCA to LCA. The activity of 7α-dehydroxylase is mainly associated with intestinal species in the genus *Clostridium*, including *C. scindens*, *C. hiranonis*, *C. hylemonae* (Clostridium cluster XVIa), and *C. sordelli* (Clostridium cluster XI) and in *Eubacterium* belonging to the *Firmicutes* phylum [189,228,229,230,231]. In rodents, dehydroxylation of C-7 in primary αMCA and βMCA results in the formation of murideoxycholic acid (MDCA) and further hyodeoxycholic acid (HDCA) [232,233,234]. Secondary BAs, ωMCA, and hyocholic acid (HCA) can also be converted to HDCA [233].

The main enzymes responsible for subsequent BAs modification include hydroxysteroid dehydrogenases (HSDH) 3α/β-HSDHs, 7α/β-HSDHs, and 12α/β-HSDHs altering groups at C-3, C-7, and C-12, respectively. The presence of the enzymes has been confirmed in a wide variety of microorganisms in GI flora, as well as free-living microorganisms including *Actinobacteria*, *Proteobacteria*, *Firmicutes*, and *Bacteroidetes* [235,236,237,238]. Coordinated activity of enzymes, usually expressed by distinct bacteria, can perform two-step epimerization consisting of oxidation of the hydroxyl group by a position-specific hydroxysteroid dehydrogenase followed by the reduction by another hydroxysteroid dehydrogenase. Therefore, UDCA is formed by 7α/β-epimerization of CDCA, and 3α/β-epimerization of CDCA results in isochenodeoxycholic acid (iCDCA). Epimerization of carbons at C-3, C-7, and C-12 positions of CA lead to the emergence of isocholic acid (iCA), ursocholic acid (UCA), and 12-epicholic acid (12-ECA), respectively. CA can also be converted to UDCA in a two-step process involving C-7α to β epimerization and 12αHSDH oxidation [228,231,236,239,240,241,242,243]. In rodents, other main metabolites of βMCA lead to ωMCA formed by 6β-epimerization and γMCA (γMCA; also known as hyocholic acid; HCA) upon 6β-epimerization and further 7β-epimerization [243,244].

A myriad of additional bacteria-catalyzed modifications include converting secondary BAs, DCA, and LCA back to CA and CDCA, DCA to LCA, and UDCA to MCA, whereas the liver has the capacity to produce HDCA from LCA [239,245,246]. In rodents, HDCA may also act as a substrate to generate ωMCA by further hepatic modification [247].

Microbial metabolism of BAs leads to increased diversity, changes in BAs’ ability to cross biological membranes via passive diffusion, Pk_a_, and hydrophobicity. Hydrophobicity is associated with enhanced binding to membrane lipids, and increased toxic and metabolic effects in the host. In general BAs’ hydrophobicity, and, therefore, also toxicity, increases in the order of ωMCA < α-MCA < β-MCA < UDCA < CA < CDCA < DCA < LCA [248]. Western-style diet, dietary, or genetic obesity all trigger gut microbiota alterations, resulting in increased levels of secondary BAs, particularly DCA. High concentrations of secondary BAs, including DCA in feces, blood, and bile, are associated with the pathogenesis of cholesterol gallstone disease, irritable bowel syndrome, and colon cancer [249,250,251,252,253,254]. DCA and LCA have been linked to colon carcinogenesis in a number of laboratory animal models and human epidemiological studies [254], with mechanisms involving selective pressure for the emergence of colonocyte mutants resistant to apoptosis (e.g., via loss of p53). Furthermore, DCA also facilitates the development of obesity-associated hepatocellular carcinoma (HCC) after exposure to chemical carcinogen in mice which could be prevented by inhibiting DCA production or antibiotics treatment [249,250]. The properties of DCA rely on its capacity to activate several carcinogenesis-related cell signaling pathways, including protein kinase C (PLC) [250], ERK1/2 via the epidermal growth factor receptor [255,256,257], β-catenin [258], JNK1/2 [259] and activation of the NOD-like receptor family pyrin domain containing 3 (NLRP3) inflammasome [260]. DCA also induces and exacerbates intestinal inflammation as well as associated dysbiosis [261,262,263]. Increased amounts of DCA in the bile of a subset of cholesterol gallstone patients correlate with the abundance of CA 7α-dehydroxylating fecal bacteria. Therapy applying antibiotics decreases the levels of fecal CA 7α-dehydroxylating bacteria, DCA in the bile, and the cholesterol saturation index in bile of these patients [264]. Interestingly, the host developed an LCA-detoxification mechanism whereby LCA acts as an activator of the vitamin D receptor, which, in turn, stimulates its metabolism and limits its toxicity [265,266]. Spectacularly, the toxic properties of DCA are applied in cosmetic surgery to adjust body shape by triggering lipolysis and fat reduction [267].

Contrary to DCA and LCA, UDCA exhibits protective effects in the gut. It prevents intestinal inflammation by promoting barrier function, inhibiting epithelial apoptosis [268], and releasing tumor necrosis factor (TNFα), interleukins IL-1β, and IL-6 [269]. UDCA also blocks bacterial growth and invasion in an *E. coli* infection model and alleviates commensal bacterial dysbiosis during the infection via the TGR5-NF-κB axis [270]. UDCA has also been approved for therapeutical use in gallstone dissolution and, due to its ability to increase BA biosynthesis, in treating primary biliary cholangitis [269,271]. Moreover, UDCA use counteracts the apoptotic and tumor-promoting effects of DCA [271,272]. The taurine conjugate of UDCA also protects against cholestasis and hepatocellular necrosis caused by more hydrophobic BAs [240,273]. However, at high doses (28–30 mg/kg/day), long-term use of UDCA leads to an increased risk of colorectal cancer in patients with ulcerative colitis and primary sclerosing cholangitis [274].

### 3.6. Deconjugation

In light of this article’s interest, another type of bacterial modification, the reaction of deconjugation and consequent freeing of taurine, is of particular interest. Facilitating the deconjugation, bile salt hydrolase (BSH) catalyzes the amide bond hydrolysis between the C-24 position of the steroid moiety and the amino acid side chain of BAs. Therefore, the absence of bacteria in GF, antibiotic-given rodents, as well as upon *Lactobacilli*-depleting tempol-treatment, result in reduced BSH activity and BA pool diversity, limiting it principally to conjugated BAs [157,275,276,277]. BSH constitutes a group of enzymes differing in subunit size, amino acid sequence, pH optimum, kinetic properties, and substrate specificity resulting in different affinity and K_m_ depending on the substrate BA. The organization and regulation of genes encoding BSH differ between species and genera. Additionally, the numbers of BSH alleles in any given strain can vary, with up to four different alleles in certain isolates [278,279]. There is evidence of horizontal transmission of BSH amongst gut bacteria suggestive of strong evolutionary selection [279]. Furthermore, both intracellular and extracellular BSHs were found. The deconjugation of BAs is mediated by all major bacterial divisions and archaeal species in resident intestinal microflora [228,279,280,281], such as *Clostridium* (*Clostridium perfringens* MCV 185, *Clostridium perfringens* 13) [282,283], Bacteroides (*Bacteroides fragilis*, *Bacteroides vulgatus*) [284,285], *Lactobacillus* (*Lactobacillus johnsonii* 100-100, *Lactobacillus plantarum* 80, *Lactobacillus acidophilus*) [286,287,288,289,290,291,292,293,294,295,296], *Bifidobacterium* [297,298,299] and *Enterococcus* [300,301]. BSH is enriched in the gut microbiota compared with other microbial ecosystems [279], but it also has been detected in *Xanthomonas maltophilia* found in soil [302,303] and *Brevibacillus* sp. isolated from hot springs [304,305].

BAs deconjugation may award a nutritional benefit as bacteria can further exploit freed amino acids as sources of carbon, nitrogen, sulfur, and energy, which has been demonstrated in *Bacteroides* [306] and was suggested in *Bifidobacterium Longum* [298]. BSH activity may as well be a mechanism of the detoxification of BAs as it is associated with increased bacterial resistance to bile toxicity [279] and an enhanced capacity to survive and colonize higher mammals’ GI tract. Accordingly, deletion of the *Bsh* gene in *Listeria monocytogenes* results in reduced resistance to BAs, gut persistence during infection, and infectivity in vivo [307,308,309]. On the one hand, the conjugated BAs occurring in protonated form may be toxic by inducing intracellular acidification [295,309]. On the other hand, deconjugated BAs possess more potent antibacterial action on *Staphylococcus aureus* [310], and successful fecal microbiota transplants in the treatment of *Clostridioides difficile* infections correlate with an increase in BSH copy number compared with levels preceding transplant, suggesting a role of BSH in protection against microbial infection [311]. BSH also facilitates the incorporation of cholesterol or BAs into bacterial membranes [312,313,314], which may increase the tensile strength of the membranes [315], change their fluidity or sensitivity to α-defensins and other host defense molecules [316,317].

Furthermore, deconjugated BAs are also known for signaling to extrahepatic tissues [189]. Therefore, the activity of BSH is consequential not only to the bacteria but also to the host. Deconjugation reduces the BAs’ efficiency in emulsifying dietary lipids, the formation of micelles, lipid digestion, absorption of fatty acids, and monoglycerides [318]. Consequently, microbial BSH activity has been linked to growth reduction in chickens [319,320], but surprisingly not in mice [287]. Efficient enterohepatic recirculation of BAs is partially dependent on their recognition in the conjugated form by active transport sites in the terminal ileum. Unconjugated BAs bind with a lower affinity to the transporters and thus may pass into the large intestine or caecum. This may result in enhanced fecal loss of BAs, which would increase the demand for cholesterol for de novo BAs synthesis that may, in turn, lower serum cholesterol levels. Accordingly, a reduction in serum cholesterol levels has been demonstrated in pigs, minipigs, GF, and conventional mice administered with BSH-active bacteria [318,321,322,323,324,325].

Importantly, deconjugation is a prerequisite for some of the enzymatic reactions, e.g., 7α/β-dehydroxylation resulting in the conversion of primary to secondary BAs [231,326,327]. However, this notion has been questioned recently [328]. Preventing BAs modification would affect the levels of ligands for various receptors. The range of BAs constitutes ligands with various affinities for their receptors, including previously mentioned FXR and TGR5 but also PXR (LCA > DCA > CA) and Vit D receptor (3-oxo-LCA > LCA > DCA > CA). Taurine-conjugated BAs are ligands for G-protein coupled receptors sphingosine-1-phosphate receptor 2 (S1PR2) and (M_2,3_) muscarinic receptors (TLCA > TDCA > TCA). Therefore, the levels of deconjugated BAs circulating from the GI tract alter BA signaling, immune and barrier function. Generation of GF mice or treatment with antibiotics or tempol, which targets *Lactobacilli*, a primary source of BSH activity in the gut, inhibits deconjugation. The absence of BSH in these mouse models reduces the production of secondary BAs serving as FXR agonists. Additionally, the levels of FXR antagonists, such as T-βMCA, accumulate, preventing CYP7A1 inhibition and increasing BA production. However, FXR-independent CYP8B1 is not affected upon BSH inhibition [157,329].

### 3.7. BAs Reconjugation

The capacity of microbiota to produce microbially conjugated BAs (MCBAs) by re-conjugation of BAs with various amino acids has been recently identified. The MCBAs are conjugated at the C-24 acyl site similarly to the host conjugation mechanism, but instead of taurine or glycine, use phenylalanine, leucine, tyrosine, valine, or leucine. Conjugation with other amino acids is likely but not yet reported. The administration of MCBAs reduces the expression of BA-synthesis genes probably through their ability to act as an agonist of FXR. In addition to rodents, MCBAs were also found in humans. Of concern, MCBAs are enriched in patients with inflammatory bowel disease or cystic fibrosis and are associated with the occurrence of bacteria *Enterocloster bolteae* [243,330,331].

## 4. Taurine in the Context of BAs and Microbiota

### 4.1. Taurine-Conjugated BAs

Taurine-conjugated BAs are generally less toxic than unconjugated BAs and exhibit some beneficial properties. Particularly, TUDCA has broad therapeutic applications. It is more efficient in treating liver cirrhosis than its deconjugated counterpart UDCA [332,333]. Its administration in mice attenuates HFD-induced hepatic steatosis, inflammatory responses, insulin resistance, and obesity. Moreover, TUDCA improves intestinal barrier function, reduces inflammatory cytokine levels and intestine histopathology scores. Finally, the gut microbiota composition of the HFD-fed and TUDCA-treated mice differs from that in HFD-fed mice, but is similar to that in chow diet-fed animals [334].

As a supplement, TUDCA is orally bioavailable, and due to its capacity to cross the blood–brain barrier, it can penetrate the central nervous system [335]. Similar to free taurine [62,63,64], TUDCA has proven neuroprotective properties which were researched in the models of Alzheimer’s disease (AD) [336,337,338,339,340,341], amyotrophic lateral sclerosis (ALS) [342], Huntington’s disease (HD) [335,343], and stroke [344,345]. In the mouse model of AD, TUDCA prevents cognitive impairments [337], interferes with amyloid-β production [336] as well as suppresses amyloid β-induced synaptic toxicity inhibiting organelle-driven apoptosis [341], and interferes with upstream molecular targets of p53 pathways [338,339,340]. In the case of ALS patients, compared with placebo treatment, TUDCA slows the progression of disability [342]. Supplementation of TUDCA in a genetic mouse model of HD reduces striatal atrophy and decreases striatal apoptosis, resulting in fewer and smaller sized ubiquitinated neuronal intranuclear huntingtin inclusions, as well as improving locomotor and sensorimotor deficits [335]. Administration of TUDCA before or up to 6 h after induction of intracerebral hemorrhage reduces apoptosis and inhibits caspase activity by 50% in the area immediately surrounding the hematoma, as well as improves neurobehavioral deficits [344].

Given that taurine supplementation increases the pool of taurine-conjugated BAs [118,119], many of the benefits of taurine mentioned in the section “Taurine” overlap with and are due to BAs activity. Furthermore, TGR5, as a BAs receptor, likely plays an essential role in these processes.

### 4.2. Taurine and Microbiota in the GI Tract

Upon release from conjugated BAs, taurine creates multiple secondary conjugates of unknown roles [132] or may be metabolized by bacteria (Figure 2). Taurine remains largely stable when cultured with human fecal samples in the absence of oxygen, although under aerobic conditions, the majority of the taurine is degraded [346]. However, another report showed that when using cat feces as the inoculum, taurine is also degraded in anaerobic cultures [347]. *E*. *coli* is an example of a bacterium that imports and utilizes taurine when cultured under aerobic conditions [348]. Therefore, the availability of taurine and its impact on the intestinal epithelium depends on whether conditions favor taurine metabolism.

Caloric restriction, which is accompanied by decreased expression of inflammatory and antibacterial genes [349], is also characterized by increased levels of taurine in the intestinal epithelium [132]. As a positive inflammasome modulator, taurine is responsible for enhanced NLRP6 inflammasome-induced IL-18 secretion upon intestinal microbial colonization, and, therefore, it modulates bacteria composition [350,351]. Consequently, taurine has been associated with inhibited growth of harmful bacteria, including *Proteobacteria* and especially *Helicobacter*, and also increasing the production of SCFA in mouse feces [351] as well as the metabolism of taurine by microbiota [352]. Controversially, some reports show that taurine does not trigger significant change in the microbiota’s diversity or composition as well as the composition of SCFA produced when cultured with human feces [346]. Reciprocally, the presence of taurine in the intestine depends on microbiota [350,353]. One of the mechanisms involves microbiota-mediated BAs deconjugation and release of free taurine [132]. Thus, microbiota transplant from calorie-restricted mice characterized by increased levels of BAs and taurine in the intestinal mucosa, raises levels of intestinal free taurine and various secondary taurine conjugates [353].

Taurine has been shown to have a substantial impact on the GI tract due to its anti-inflammatory properties and, therefore, impacting the environment of resident gut bacteria. When supplemented in the model of an immunosuppressive mouse, taurine improves immune cell numbers in Payer’s patches [354]. It also attenuates dextran sulfate sodium (DSS)-induced colitis reducing the severity of diarrhea, rectal bleeding, colon shortening, histological score, myeloperoxidase (MPO) activity elevation, abnormal macrophage inflammatory protein (MIP-2) gene expression, and infiltration of neutrophils [355,356]. Consequently, it retards DSS-induced weight loss and lowers mortality upon DSS-induced treatment [350,355]. In the model of trinitrobenzene sulfonic acid (TNBS)-induced inflammatory bowel disease, taurine reduces the inflammatory parameters in rat colon by increasing capacity to defend against oxidative damage [357]. Taurine also inhibits the TNF-α-induced secretion of IL-8 from human intestinal epithelial Caco-2 cells [355]. Particularly noteworthy is the role of taurine in T lymphocytes, where it accounts for approximately 44% of the total free amino acid pool [358] and is critical for cell survival, T cell-mediated immune reactions, and memory development [359]. However, recently an extraordinary mechanism preventing infection has been described in which BA-derived taurine mediates long-term metaorganism colonization resistance (Table 1). Mechanistically, the host, triggered by transient intestinal infection, alters BAs metabolism and deploys taurine leading to the expansion of taurine-utilizing taxa. Taurine nourishes and trains the selected microbiota. Following infection, *Deltaproteobacteria*, typically a minor class, expands up to 100-fold. Importantly, *Deltaproteobacteria* encompasses sulfate-reducing bacteria possessing the capacity to convert taurine to ammonia, acetate, and sulfide. Subsequently, sulfide serves as an inhibitor of cellular respiration, which is key to host invasion by numerous pathogens. The conversion of taurine occurs via taurine–pyruvate aminotransferase encoded by the tpa gene, which is more prevalent in infection-trained microbiota metagenomes. Following oral gavage with *Klebsiella pneumoniae*, the subsequent fecal transfer of *Klebsiella pneumoniae* infection-trained microbiota enhanced resistance to colonization. *Klebsiella pneumoniae* grows on 1,2-propanediol by scavenging oxygen via cytochrome oxidase bd-II. Sulfide at concentration >250 mM potently inhibited *Klebsiella pneumoniae* respiration and ability to grow on 1,2- propanediol with oxygen. Notably, supplying exogenous taurine alone was sufficient to induce alteration in microbiota function and enhance resistance. However, taurine treatment of GF mice did not enhance resistance to *Klebsiella pneumoniae* [360]. Previously, TCA has also been implied to stimulate intestinal bacteria capable of converting taurine to hydrogen sulfide [352]. Sadly, bismuth subsalicylate, a common over-the-counter drug for diarrhea and upset stomach, neutralizes infection protection as it inhibits hydrogen sulfide production [360].

In addition to being toxic to bacteria, sulfide can also be toxic to the host. Sulfide increases proliferation in the intestinal crypt, epithelial cells, and the upper colonic crypts, accompanied by induction of inflammatory pathways [361,362,363]. Furthermore, sulfide, as a genotoxic compound, triggers oxidative stress leading to cell-cycle arrest and DNA damage in the human colon [362,363,364,365]. Interestingly, in the adenocarcinoma cell line HCT116, sulfide was implicated in preventing apoptosis induced by β-phenylethyl isothiocyanate, a phytochemical found in cruciferous vegetables [366].

Endogenous concentrations of sulfide range between 0.2–3.4 mmol/L in the GI tract of mice and humans [367,368]. Sulfide-detoxifying enzymes are upregulated during differentiation in the human colon [369], and rat colonocytes express protective enzymes on the mucosal surface that oxidize sulfide [370]. However, these enzymes are decreased in the colon of patients suffering from cancer and active ulcerative colitis [369]. Accordingly, fecal sulfide is significantly elevated in ulcerative colitis patients experiencing active disease [361,371,372]. Controversially, some reports have shown no significant increase in sulfide in ulcerative colitis patients [373].

Concerning microbiota, sulfidogenic bacterium (*Fusobacterium* spp.) is associated with the tumor surface in a subset of colorectal cancer [374,375], while sulfite-reducing opportunistic pathogen *Bilophila wadsworthia* bacteria, which is difficult to detect in healthy individuals, emerges under pathological conditions such as appendicitis [376], ulcerative colitis [377,378], and colorectal cancer [379]. *B. wadsworthia* thrives in the presence of taurine-conjugated BAs since it utilizes taurine-derived organic sulfur as the terminal electron acceptor of the electron transport chain resulting in the generation of sulfide as a byproduct [380,381]. Increased taurine conjugation of hepatic BAs, e.g., by consuming a diet high in saturated (milk-derived) fat, breeds *B. wadsworthia*. What is relevant in this context, milk fat promotes the onset and incidence of colitis in IL-10^−/−^ mice, driving it from a spontaneous rate of 25–30% to over 60% in a 6-month period. Correspondingly, *B. wadsworthia* occurrence, as well as TCA supplementation, are also associated with colitis development in IL-10^−/−^ mice. The mechanisms involve *B*. *wadsworthia*-driven activation of dendritic cells in a way that selectively induces the production of interferon-γ (IFNγ) and Th1- mediated colitis [120]. A diet high in meat has been shown to significantly increase both the levels of taurine conjugation to BAs [118,119] and the production of hydrogen sulfide in the colon [367]. The low occurrence rate of cancer in African populations consuming small amounts of meat is associated with colonic bacteria fermentation [382]. However, native black Africans also have decreased ratios of taurine to glycine conjugation (1:9) and low hydrogen sulfide production compared with populations consuming a “Western diet”. Therefore, consumption of a diet low in taurine may contribute to the reduced frequency of cancer. Opportunely, supplementation with ω-3 fish oil inhibits the bloom of *B*. *wadsworthia*, most likely because of alterations in the BA composition [383].

In addition to its toxic properties, sulfide and sulfate prebiotics also stimulate GLP-1 secretion and its downstream metabolic actions [384]. Furthermore, supplementation of prebiotic chondroitin sulfate to mice increases the proportion of *Desulfovibrio piger*, a sulfate-reducing bacterium, and accentuates GLP-1 levels, leading to an improved glucose tolerance [384]. In contrast, another cell-based study revealed that sulfide has potent inhibitory effects on TGR5-mediated GLP-1 and PYY release [385]. Thus, the effect of hydrogen sulfide on GLP-1 release remains controversial and calls for further investigation.

### 4.3. Taurine, Microbiota, and Cardiovascular Diseases

Gut microbiota is a critical risk factor in cardiovascular diseases as it impacts host metabolism and immune homeostasis. Depletion of the gut microbiota by antibiotics has been shown to reduce the incidence of intracranial aneurysms in mice [386]. Similarly, taurine plays a protective role in acute ischemic stroke [387], subarachnoid hemorrhage [388], and aortic aneurysm formation [389].

The microbiome of atherosclerotic cardiovascular disease is characterized by depletion of the taurine transport system [390]. Importantly, taurine depletion is a key factor in the pathogenesis of unruptured intracranial aneurysms (UIA) [391]. Microbiota transplantation from UIA patients’ donors is sufficient to induce UIAs and decrease the serum taurine levels of mice, indicating that UIA microbiota mediates the low level of taurine in mice. Specifically, the abundance of *Hungatella hathewayi* is strikingly reduced in UIA and correlates positively with the circulating taurine concentration in humans and mice. Consequently, gavage with *H. hathewayi* normalizes the taurine serum levels and protects mice against the formation and rupture of intracranial aneurysms [391]. *H. hathewayi* also reduces the release of cytokines and lowers NF-κB activation in dendritic cells [392], whereas taurine supplementation reverses the progression of intracranial aneurysms [391].

### 4.4. Taurine and GSH

Taurine and GSH are linked on several levels. Both require cysteine for their synthesis, therefore, the responsible enzymes may compete for the substrate [393]. Both act as antioxidants and play a vital function in mitochondria. GSH serves as a mitochondrial redox buffer to stabilize the electrical gradient, whereas taurine is applied as a pH buffer, but simultaneously establishes the equilibrium between the NADH/NAD+ redox pair and the redox buffer pair GSH/GSSG [394]. Taurine enhances the activity of antioxidant enzymes, including SOD, CAT, GSH peroxidase (GPx), and GSH reductase (GR), thus preserving redox levels and GSH stores [76,395,396,397]. Upon one-time taurine supplementation in rats, GPx activity shows an increase in liver, heart, stomach, and plasma; GR activity increased in the kidney and decreased in liver and plasma, whereas GSH levels increased in the liver and stomach and decreased in the kidney [395]. Therefore, in several instances, taurine has been found to prevent or repair oxidative damage by acting on GSH. Following nicotine administration, taurine protects against oxidative stress by normalizing GSH stores in rats [398]. It reduces oxidative stress in iron-overloaded mice and protects the levels of reduced GSH [399]. Further, taurine treatment alleviates adverse effects of mitochondrial oxidative stress found in induced pluripotent stem cells (iPSCs) from a patient with mitochondrial myopathy, encephalopathy, lactic acidosis, and stroke-like episodes (MELAS) by normalizing stores and the ratio of the reduced to oxidized GSH (GSSG) [400]. Finally, taurine administration improves both DNA damage and oxidative indices triggered by acetaminophen. In this case, taurine was shown to act by reducing MDA formation, increasing the activity of antioxidant enzymes, and regulating synthesis, utilization, and reduction in GSH [401].

Gut microbiota influences both host taurine [157] and GSH metabolism [402]. Recently, we have shown that upon deconjugation from BAs, taurine creates various conjugates, among others, with GSH. The occurrence of the conjugate in the intestinal epithelium and activity of GSH S-transferases (GST) catalyzing the reaction is modulated by microbiota. This takes place, likely, by bacterial BSH activity regulating taurine availability. The conjugation of taurine with GSH increases intestinal taurine uptake during caloric restriction, playing an important role in taurine circulation and reuse. However, the potential roles of other taurine conjugates remain unknown [353,403].

### 4.5. Taurine and Microbiota in Autism

Autism is associated with frequent dietary issues and GI symptoms, including abdominal pain, constipation, diarrhea, gastroesophageal reflux, bloody stools, vomiting, and gaseousness [404,405,406]. The symptoms correlate with the severity of core autism-related behavioral abnormalities on measures of irritability, anxiety, and social withdrawal [407]. Increased intestinal permeability, which is connected with the potential for translocation of intestinal metabolites or bacteria and consequent immune activation, is linked to autism [404,408,409]. Dysregulated GI motility and secretion in autistic individuals are connected with an altered composition of intestinal microbiota, dysbiosis. The changes in the GI tract influence higher-order behavioral and brain function via the gut–brain axis, vagus nerve, indirect immune and metabolic signals [410,411,412,413].

In one study, taurine serum concentrations in children with autism spectrum disorder (ASD) were not significantly different from their parents or siblings; however, 21 out of 66 children with ASD had low taurine concentrations, which may have consequences on their mitochondria function. Accordingly, lowered taurine levels were proposed as a biomarker of autism [414]. Transplanting gut microbiota from human donors with ASD into GF mice induces hallmark autistic behaviors. The brains of mice colonized with ASD microbiota display alternative splicing of ASD-relevant genes [415]. The circulating concentration of taurine is ~50% reduced in GF mice receiving fecal transplantation from individuals with an ASD. Furthermore, the metabolism of various amino acids, specifically that of proline, taurine, glutamate, and glutamine, are differentially represented in the metagenomes of mice receiving ASD-microbiota. Bioinformatical predictions imply that taurine concentrations might result from differential synthesis potential by three species: *Alistipes* sp. HGB5, *Alistipes finegoldii*, and *Bacteroides xylanisolvens*, whereas taurine supplementation improves repetitive and social behaviors and reduces anxiety in mice by acting locally in the gut [415].

### 4.6. Taurine and Microbiota in Non-GI Tissues

Importantly, due to its role in preventing infections, taurine interacts with bacteria also in tissues other than the intestine. The infection of mammary epithelial cells with *Streptococcus uberis* is connected with the internalization of the pathogen, thus leading to avoiding the elimination of bacteria by medication and host responses. Taurine attenuates the infection via phosphoinositides/Ca^2+^ signaling, inhibition of over-activation of the NF-κB pathway, and stimulation of Treg cells [416,417,418]. It also activates autophagy via phosphatase and tensin homolog (PTEN) and Akt/mTOR, which accelerates the degradation of intracellular *S. uberis*, reduces intracellular bacterial load, and alleviates the inflammation and damage caused by the infection [419].

Both taurine and microbiota play multiple roles in organs outside of the GI tract as well as in response to various diseases. Due to its anti-inflammatory and anti-oxidative properties, taurine alleviates liver injury and its consequent events, including a rise in plasma and brain ammonia and brain oedema [420,421]. It also prevents liver steatosis by reducing oxidative damage, inhibiting lipogenesis, and promoting energy expenditure [421]. Importantly, liver diseases are tightly connected with dysbiosis, and taurine has been suggested to prevent hepatic inflammation by inhibiting TLR4/MyD88 [68,422,423]. TLR4 recognizes the pathogen-associated molecular pattern and allows the host to identify microorganisms, ultimately transmitting bacterial signals the play a pivotal role in the gut-liver axis [423]. Thus, taurine may influence the bacteria signaling to extragastrointestinal tissues.

In the renal system, taurine is particularly important for osmoregulation. However, it also reduces the injurious effect of several kidney diseases, including diabetic nephropathy, glomerulonephritis, chronic renal failure, and acute kidney injury [424,425,426,427]. Similarly, intestinal flora has been reported to prevent the development and progression of several renal diseases, such as lupus nephritis, chronic kidney disease, diabetic nephropathy, and renal ischemia–reperfusion injury [428]. Similarly, both taurine and gut bacteria play a role in cardiovascular diseases [429,430], neurological [51,53,54,55,56,57,58,60,61,62,63,64,431] and multiple other disorders. Their activity is very likely coordinated; however, so far, it lacks evidence.

**Table 1 cells-11-02337-t001:** Direct evidence on the physiological consequences of the interaction of BAs, taurine, and microbiota or taurine and bacteria.

Summary of the Results	Reference
BAs, taurine, and microbiota	
*Deltaproteobacteria* metabolizes BAs-derived taurine to sulfide, which serves as a mechanism to prevent infections	[360]
*B. wadsworthia* metabolizes BAs-derived taurine to sulfide, which triggers colitis	[120]
Microbiota releases taurine from BAs, leading to the creation of taurine-GSH conjugates and an increase in taurine uptake	[353]
Taurine and bacteria	
*H. hathewayi* normalizes the taurine serum levels and protects mice against the formation and rupture of intracranial aneurysms	[391]
Reduced concentration of taurine in individuals with autism spectrum disorder (ASD) rely on gut bacteria	[415]
Taurine affects liver health by regulating bacterial signals transferred through TLR4/MyD88	[68]
Taurine attenuates the infection of mammary epithelial cells with *Streptococcus uberis*	[416,417,418,419]

## 5. Conclusions

Taurine acting as a conjugate of BAs influences the signaling of TGR5 and FXR, having serious physiological consequences. Upon deconjugation, it dynamically interacts with microbiota by being metabolized or leading to the production of multiple signaling molecules. Alternatively, it is taken up to fulfill its role in other tissues. The crucial reaction of taurine release relies on microbiota, and the sensitive equilibrium and interaction of factors BAs, taurine, and microbiota contribute to the health and homeostasis of the host.

## Figures and Tables

**Figure 1 cells-11-02337-f001:**
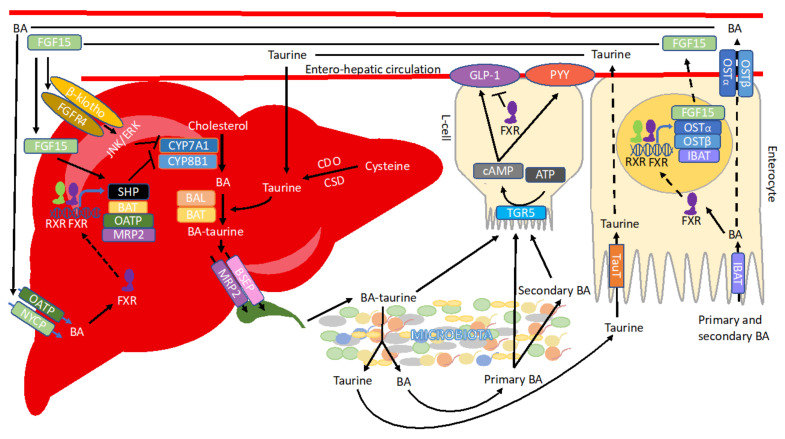
**Regulation of bile acids** (**BA**) **metabolism.** BAs are produced in the liver by modifying cholesterol with a series of CYP enzymes. The liver also produces taurine, which can be conjugated to BAs by BAL (BA CoA-ligase) and BAT (BA CoA:amino acid N-acyltransferase) enzymes. BAs secreted from the liver into the intestine undergo deconjugation and a series of modifications generating an array of secondary BAs. The deconjugated taurine is taken up by TauT and BAs by IBAT (ileal BA transporter) and OSTα-OSTβ (organic solute transporter α-β). Afterward, BAs are recirculated via entero-hepatic circulation and transported into the liver by NTCP (sodium-dependent taurocholate co-transporting peptide) and OATP (organic anion-transporting polypeptides). The presence of BAs activates their receptors. Upon stimulation, intestinal TGR5 promotes GLP-1 (glucagon-like-peptide-1) and PYY (peptide-YY) production. At the same time, nuclear receptor FXR (farnesoid X receptor) regulates the expression of genes connected with BAs transport and signaling. One of FXR target proteins, FGF15 (fibroblast growth factor 15), transfers the signal of BAs abundance from the intestine to the liver. Consequently, it reduces BAs’ production and transport via JNK/ERK signaling pathway or together with SHP (small heterodimer partner). Additionally, hepatic FXR surveys for the levels of BAs and signals to adjust BAs’ biosynthesis, conjugation, and transport.

**Figure 2 cells-11-02337-f002:**
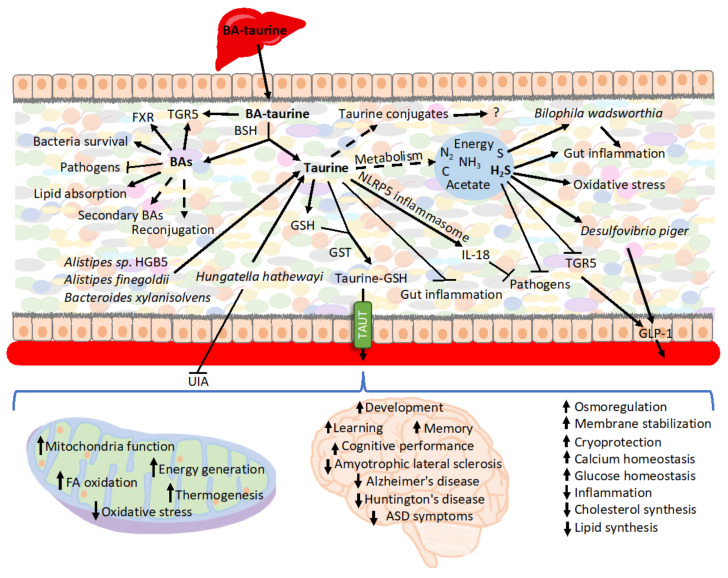
**Summary of the roles of BAs-derived taurine.** Various bacterial strains have the capacity to deconjugate BAs. The released unconjugated BAs modulate gut microbiota composition, signal various functions through its receptors, and impact nutrient uptake. BAs are also submitted modifications by microbiota, including reconjugation and generation of secondary BAs. The faith of taurine released from conjugated BAs in the intestine can follow various paths. It is metabolized for energy and generates secondary metabolites, which, such as H_2_S, may play a role in interacting with bacteria, inflammation, and oxidative stress. Taurine also signals within the intestine to extinguish inflammation and prevent pathogens colonization. Additionally, various compounds can conjugate taurine, and, e.g., conjugation to GSH enhances taurine uptake during caloric restriction. The exported taurine plays various roles in other organs, particularly in the nerve system as well as in mitochondria all over the body.

## Data Availability

Not applicable.

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
