# Peer review of "Versatile Triad Alliance: Bile Acid, Taurine and Microbiota"

_cells, 2022, doi:10.3390/cells11152337_

Round 1
Reviewer 1 Report
Dear author,
The article was carefully revised and suggested moderate revisions.

Author Response
First of all, I would like to thank for the detailed revision of the article and your efforts to improve the manuscript.
- Please avoid using long sentences.
Sentences have been split in shorter in several instances along the text.
2. Please submit a plagiarism report.
Please find attached the plagiarism report.
3. Indicate the sources of Taurine in fruits and, vegetables and in medicinal plants
Information concerning taurine content in vegetables, nuts or seeds, fruits, legumes, and medicinal plants has been introduced in section 2. However, I managed to find only one example of medicinal plant that contains substantial levels of taurine.
4. A briefly highlights the role of Taurine in reproduction in section” 2. Taurine”
The role of taurine in the introduction is now mentioned in section “2. Taurine”
5. Figure-1 description is required.
The description for figure 1 has been added.
6. Again Figure-2 description is required.
The description for figure 2 has been added.
7. The author only focused on Taurine, microbiota relationship with cardiovascular diseases, autism, non- GI tissues. There is a need to explore more functions of this relationship in other health disorders, like intestinal diseases, hepatic problems and the urogenital system.
The last section of the manuscript now contains appropriate information. However, I could find very few articles proving the direct connection between microbiota, taurine, and diseases. If I missed any of the important articles, please inform me.
8. There are some minor mistakes in English, that needs to be corrected.
Language has been corrected.
9. Include tables of previous studies conducted on this aspect.
It was not clear to me what you mean by “this aspect”. I assumed I should make a table listing direct connections between bile acids, microbiota, and taurine. There are very few articles proving the direct connection, and to create the full picture of each aspect, I had to cite multiple articles contributing to the idea. As the table for bile acids, microbiota, and taurine would be very short, I created a subsection: microbiota and taurine. I hope that you find the results in table 1 satisfying.
10. Incorporate cellular evidence which makes the worth of review.
I am sorry, but this comment was not clear to me.
11. Clinical studies on this aspect and their outcome is missing, including that one.
Again, I am sorry, but this comment was not clear to me. What do you mean by “this aspect” and “that one”.
Reviewer 2 Report
The manuscript by Duszka is an interesting and well argumented review on the role of taurine and its interactions with bile acids and mirobiota
The manuscript is well arranged and contains useful information for the readers.
As a minor comment, both figures could be accompanied by a brief description to facilitate their interpretation.
Author Response
Thank you very much for the positive evaluation.
Legend has now been added to both figures.
Round 2
Reviewer 1 Report
Dear authors,
Please submit a plagiarism report.